# Fractal Cracking Patterns in Concretes Exposed to Sulfate Attack

**DOI:** 10.3390/ma12142338

**Published:** 2019-07-23

**Authors:** Jinwei Yao, Jiankang Chen, Chunsheng Lu

**Affiliations:** 1Key Laboratory of Impact and Safety Engineering, School of Mechanical Engineering and Mechanics, Ningbo University, Ningbo 315211, China; 2Zhejiang Business Technology Institute, Ningbo 315012, China; 3School of Civil and Mechanical Engineering, Curtin University, Perth, WA 6845, Australia

**Keywords:** concrete, sulfate attack, micro-crack propagation, digital image processing, fractal dimension

## Abstract

Sulfate attack tests were performed on concrete samples with three water-to-cement ratios, and micro-crack growth patterns on concrete surfaces were recorded. The expansive stress and crack nucleation caused by delayed ettringite formation (DEF) were studied using X-ray diffraction and scanning electron microscopy. By means of a digital image processing technology, fractal dimensions of surface cracking patterns were determined, which monotonously increase during corrosion. Moreover, it is shown that the change of fractal dimensions is directly proportional to accumulation of DEF, and therefore, a simple theoretical model could be proposed to describe the micro-crack evolution in concretes under sulfate attack.

## 1. Introduction

Corrosion damage in concretes is an important influential factor on the durability of concrete structures in a marine environment (e.g., [1,2,3,4]). Although much effort has been devoted to investigating sulfate attack in concretes, there is still a lack of full understanding on its physical mechanism because it is a complicated physicochemical destructive process [5,6,7,8,9]. For example, Boyd and Mindess [10] studied compressive and tensile strengths of several types of concretes and showed that, when concrete is exposed to a sulfate corrosive environment, the water-to-cement ratio should be taken into account for verification of its durability. Moreover, the mix proportion of concretes is a key parameter for their resistance to damage. Based on the damage evolution in concretes subjected to sulfate attack, Shanahan and Zayed [11] found that the changes of material expansion and compressive strength can be used to evaluate the concrete durability under corrosion attack.

With the aid of scanning electron microscopy (SEM) and X-ray diffraction (XRD), Chen et al. [12] conducted an experimental study on expansion and destruction of concretes immersed in sulfate solution with different concentrations, and identified internal expansive forces generated by delayed ettringite formation (DEF) and gypsum as the main reason for expansion. Moreover, they established a theoretical prediction model for stress, strain and volume expansion during sulfate attack. Müllauer et al. [13] measured the expansion rate of mortar samples as a function of corrosion time, and then studied the chemical composition of the corrosive material by XRD. It was shown that sulfate attack damage was due to stress formed by DEF ranging from 10 to 50 nm, which exceeds the tensile strength of a cement matrix by 8 MPa. Furthermore, they found that by increasing the sulfate concentration and content of tricalcium aluminate, the material was more easily destroyed at the high stress level.

In the process of sulfate corrosion in concretes, generation of DEF and gypsum may cause internal expansion forces [14,15,16]. Under the sulfate attack condition, DEF in concretes is actually a complicated chemical reaction process and the damage evolution of concrete is mainly represented by concrete expansion, crack nucleation and propagation. The inherent damage evolution mechanisms involved in corrosion of concretes can be determined only after a detailed understanding of the involved chemical reactions, micro-crack nucleation and propagation in concrete, as well as the quantitative interrelationship between the crack groups caused by corrosion damage in concrete and chemical reactions. However, due to the high complexity of cracking in concrete, this is usually difficult to accomplish, if not impossible, by conventional fracture theories [17]. However, the concept of fractal provides us with a powerful tool for describing irregular and segmented graphics [18,19,20,21,22,23].

Over the past few decades, numerous studies have focused on the measurement of crack patterns in the field scale. With the development of computers, there is an increasing opportunity to analyze data collected by using image processing techniques because of their advantages of fast, accurate and non-destructive quantification [24,25,26,27,28]. This also promotes wide applications of the fractal theory [29,30,31], especially in the research field of concretes [32,33,34,35]. For instance, Mechtcherine [36] performed a uniaxial tensile test on a concrete specimen and analyzed the fractured surface by fractal dimension. They found that the crack types and roughness of fractured surfaces were closely related to heterogeneity of concrete. Carpinteri and Brighenti [37] used the fractal theory to study the fracture behavior of plain and fiber-reinforced concretes with different water-to-cement ratios. It was shown that fracture energy can be characterized by fractal dimensions and fracture resistance of fiber-reinforced concrete was verified. Erdem and Blankson [38] analyzed the characteristics of concrete fractured surfaces under dynamic loading by digital image processing and laser scanning technology. They reported that the higher the fractal dimension was, the greater the fracture energy that was required at the interface. Szeląg [39,40] suggested that conventional optical scanners can be applied to scan samples with a very high resolution, and then they detected cracks on material surfaces that cannot be seen by the naked eye. In addition, a digital image processing technology could also be used to process surface crack images. Thus, this provides another effective method for quantitative evaluation of a crack propagation law [41,42].

In this paper, surface cracking of concrete specimens with different water-to-cement ratios was studied when they were exposed to environments of sulfate attack and wetting–drying cycles. The digital image processing technique was used to monitor the concrete surface crack propagation. The characteristics of concrete surface cracks and their corresponding damage degree were obtained. A box-counting method was used to analyze the fractal dimension of surface cracks in concrete specimens. Finally, the relationship between the fractal dimension of surface cracks and corrosion time was discussed from the perspective of chemical reaction rate equations. A fractal evolution model of micro-cracks in concretes under sulfate attack was established, which can better describe the characteristics of damage evolution.

## 2. Experimental Methodology

### 2.1. Materials

In tests, P·C 32.5R composite Portland cement produced by Ningbo Conch Cement Co. Ltd (Ningbo, China) was used. Its chemical composition was determined by an X-ray fluorescence spectrometer, as listed in Table 1. The mineral composition of cement was obtained by quantitative XRD, and the relevant results in Table 2 met requirements of the code “GB 12958-1999 Composite Portland Cement”. The fineness modulus of sand was 2.65, which was consistent with that of the code “GB/T 14684-2011 Construction Sand”. The aggregate was a continuous grade of gravel with the grain size between 5 and 10 mm. The particle size distributions of sand and aggregate used are shown in Figure 1. The local tap water in Ningbo was used for preparation of concrete specimens and soaking solution. Sulfate used in tests was the anhydrous sodium sulfate analytical reagent produced by Jiangsu Qiangsheng Functional Chemical Co. Ltd (Changshu, China), with a concentration of more than 99%.

### 2.2. Mixture Design and Sample Preparation

In this study, three mixture ratios were considered, corresponding to water-to-cement ratios of 0.45, 0.55 and 0.65. Here, it is worth noting that in practical engineering, concrete with w/c = 0.65 is rarely used. However, in order to clearly determine the influence of w/c on the crack evolution and its fractal dimension, such a water-to-cement ratio was chosen [40]. The weight ratio of cement, sand and gravel in concrete samples was 1:2:2. The sample size was 60 × 60 × 20 mm^3^. First, samples were molded and naturally cured for 24 h. Then, these samples were placed in a curing box with a temperature of 20 ± 2 °C and relative humidity of 95% for 28 days. Finally, the samples were placed in 8% sulfate etching solution for a dry-and-wet cycle in an indoor environment. The object studied in this paper is surface crack, and the surface of the sample is mainly cement paste, as shown in Figure 2. It should be pointed out that the aggregates would affect the nucleation and growth of corrosion cracks. However, from Figure 2 we can see that only a few aggregates appear, and most of the area of the surface is covered by cement paste. This implies that nucleation and propagation of corrosion cracks dominantly occur in the cement past. Therefore, the effect of aggregates on the evolution of corrosion cracks could be approximately neglected.

### 2.3. Method

Concrete samples were first immersed in sodium sulfate solution of 8 wt.% for 1 day (as the wet state). Then the samples were taken out from sodium sulfate solution and dried naturally for 1 day. The total process was considered as a wetting–drying cycle. In order to keep the constant concentration of solution during soaking and prevent evaporation, the solution container was sealed with a plastic lid. Solution was periodically changed every month to ensure the solution composition. The corrosion tests were carried out indoors to keep a constant temperature and humidity. To avoid the effect of temperature on deterioration of concretes during drying, samples were naturally dried at room temperature. Taking their nucleation times and propagation rates of surface cracks into account, the evolution of surface cracks in samples with different water-to-cement ratios were assessed in various time intervals of corrosion, as shown in Figure 3.

During tests, it was observed that concrete samples with a water-to-cement ratio of 0.65 were initially cracked after around 90 days and were completely damaged after 250 days. Whereas, initial cracks in samples with a water-to-cement ratio of 0.55 were observed after around 180 days and they completely broke after 360 days. The initial crack and full damage of samples with a water-to-cement ratio of 0.45 were recorded after around 250 and 630 days, respectively. Although cracking patterns (including the length and direction distributions of micro-cracks) formed in concretes with three water-to-cement ratios are obviously different, micro-cracks in all samples propagate from the periphery to the center.

## 3. Fractal Characteristics of Surface Cracks

### 3.1. Image Processing of Surface Cracks

To monitor the surface cracks in concrete specimens during corrosion, a SONY DSC-TX30 digital camera, (Sony China Corporate, Shenzhen, China), was installed on a fixed platform. The camera with a resolution of 18.20 mega pixels and an optical zoom of 5 × was adjusted 0.3 m directly above a sample. To ensure the same geometric distortion for all samples, the shooting square area of 60 × 60 mm^2^ was determined by the projection of the camera lens.

Based on Ren et al. [43] and an open image processing software (ImageJ ver. 1.8.0) (National Institutes of Health, Bethesda, MD, USA), digital image processing on surface cracking patterns of a concrete specimen under sulfate attack is illustrated in Figure 4. First, a polychromatic method was used to geometrically correct the color photo of surface cracks. After calibrating the image scale, image pixels were converted into actual length (in millimeters) and the image was cropped (Figure 4a). Next, color images of each sample were converted into grayscale ones and the latter were binarized using a grayscale threshold. The grayscale images were reversed (Figure 4b), and in this way, cracking areas of each sample were white regions on images. Then, in the case of a given threshold of crack length, a conventional opening operation was performed to reduce small white spots on aggregates, which were considered to be noise in binary images. If this was not effective, the remaining white spots (possibly represented aggregates) in the non-cracked area were removed by manual intervention (Figure 4c). Further, in the case of a given threshold of crack length, a crack fusion method was proposed to repair the crack. The binary image was expanded to eliminate a narrow space where the crack spacing was less than a threshold and the pixels were removed from the crack boundary. The skeletonization algorithm was then performed until a pixel wide crack skeleton was obtained (Figure 4d). Finally, a certain threshold was considered to tailor cracks with lengths less than the threshold (Figure 4e) as noise generated during skeletonization [26].

On this basis, pixels were counted from the two-dimensional skeleton of a crack pattern to determine the crack length. Here it is worth noting that in this study, different thresholds were considered for concrete samples to improve accuracy. However, in image processing, the batch pattern could also be used to extract the crack features under the same threshold, but the extracted results might be less accurate.

### 3.2. Fractal Dimension of Surface Cracks

The box-counting method [44,45] was used to calculate the fractal dimension of surface cracks. The surface cracking pattern was covered by a square box with a side length of 1 and then the square box was divided into a grid set containing (1/*r*)^2^ small square boxes with a side length of *r* (*r* = 1, 1/2, 1/4, 1/8, 1/16, 1/32 and 1/64), as shown in Figure 5. Covering these cracks with such a grid set and counting the number of boxes *N*(*r*) with cracks in the grid at each scale, the fractal dimension (*D*) of the surface cracking pattern can be determined by
(1)log[N(r)]=−Dlog(r)+const.
as shown in Figure 6 for three samples with different water-to-cement ratios.

It is obvious to see in Figure 7 that the fractal dimensions of surface cracks in all these samples monotonically increase with increasing corrosion time. Among them, the reason why some fractal dimensions *D* are less than 1 (for example, *w*/*c* = 0.45, *t* = 120 days and 140 days) is that the surface crack of the sample has only just appeared, and the total length of the crack is very short. Compared with the surface size of the sample, it does not have the properties of lines, but only points or short lines.

### 3.3. Damage Degree Caused by Surface Cracks

Let us consider the case as shown in Figure 5f, where the surface of a concrete sample was covered with a grid set of *r* = 1/64. The damage degree (%) caused by surface cracks can be defined as *d* = *N*/4096 × 100%, with *N* the total number of surface cracks passing through a small square box. It is seen in Figure 8 that the damage degree of surface cracks monotonically increases with the increase of corrosion time. Similarly, the damage degree increases by increasing the water-to-cement ratio. The results indicate that the water-to-cement ratio of concretes, as an important factor of corrosive solution, can considerably influence the damage degree.

### 3.4. Relationship between Damage Degree and Fractal Dimension

As shown in Figure 9, there is a power law relationship between the fractal dimension of surface cracks and its corresponding surface damage degree in a concrete specimen. However, the relationship seems independent of the water-to-cement ratio of concretes. That is, such a relationship can be considered to be universal.

## 4. Theoretical Damage Model

In order to establish a theoretical model for damage evolution, the composition of concretes and their crystal structures and morphologies were obtained by XRD, as shown in Figure 10. Moreover, microscopic observation was carried out by SEM and energy spectrum analysis was made to determine the composition of material elements. Figure 11 shows the microscopic morphology and associated chemical composition of DEF.

The experimental results show that DEF in concrete pores is the main cause of corrosion damage. The diffused sulfate ions in concrete pores react with the pore solution to form DEF and result in damage initiation. Experimental results indicate that the surface cracks are different for specimens with different water-to-cement ratios (*w*/*c*). Because different water-to-cement ratios (*w*/*c*) leads to different pore structures in specimens, the higher the value of *w*/*c*, the more the surface cracks due to higher porosity. Otherwise, the nucleation of corrosion cracks occurs on the corner of specimens, which is because three-dimensional diffusion takes place on the corner.

The process could be divided into the two steps of (1) sulfate ion (${\mathrm{SO}}_{4}^{2-}$) reacts with calcium hydroxide (CH) to form gypsum ($C\bar{S}H{}_{2}$); and (2) the formed gypsum reacts with tricalcium aluminate (C_3_A) and water (H) to form DEF ($C_{6}A{\bar{S}}_{3}H_{32}$). The detailed chemical reaction equations can be summarized as [46,47]:(2)$$\mathrm{CH}+{\mathrm{SO}}_{4}^{2-}\to C\bar{S}H_{2}+2{\mathrm{OH}}^{-}$$
(3)$$\begin{array}{l} \mathrm{CA}+qC\bar{S}H_{2}\to C_{6}A{\bar{S}}_{3}H_{32}\\ \mathrm{CA}=\gamma_{1}C_{3}A+\gamma_{2}C_{4}{\mathrm{AH}}_{13}+\gamma_{3}C_{4}A\bar{S}H_{12}\\ q=3\gamma_{1}+3\gamma_{2}+2\gamma_{3} \end{array}$$
where $C_{4}{\mathrm{AH}}_{13}$ and $C_{4}A\bar{S}H_{12}$ represent hydrated C_3_A and mono-sulfate, *q* is the stoichiometric weighting factor of the sulfate phase, and γ_*i*_ (*i* = 1−3) indicates the ratio of each aluminate phase to the total aluminum content.

It is seen from Equations (2) and (3) that the change of expansive stress depends on the increase rate of DEF and the latter is related to the rate of the chemical reaction. Here, the chemical reaction rate control equations are
(4)$$\frac{dC_{\mathrm{SO}}}{dt}=-kC_{\mathrm{SO}}C_{\mathrm{CA}}$$
(5)$$\frac{dC_{\mathrm{CA}}}{dt}=-\frac{k}{q}C_{\mathrm{SO}}C_{\mathrm{CA}}$$
(6)$$\frac{dC_{\mathrm{DEF}}}{dt}=\frac{k}{q}C_{\mathrm{SO}}C_{\mathrm{CA}}$$
where *k* is the chemical reaction constant, $C_{\mathrm{SO}}$ is the concentration of sulfate ions, $C_{\mathrm{CA}}$ is the concentration of C_3_A, and $C_{\mathrm{DEF}}$ is the concentration of DEF.

The initial conditions for these equations are as follows:(7)$$\begin{array}{l} {C_{\mathrm{SO}}|}_{t=0}=C_{{\mathrm{SO}}_{0}}\\ {C_{\mathrm{CA}}|}_{t=0}=C_{{\mathrm{CA}}_{0}}\\ {C_{\mathrm{DEF}}|}_{t=0}=C_{{\mathrm{DEF}}_{0}} \end{array}$$

By combing Equations (4) and (5), we have
(8)$$\frac{dC_{\mathrm{CA}}}{dt}=-kC_{\mathrm{CA}}^{2}+(kC_{{\mathrm{CA}}_{0}}-\frac{k}{q}C_{{\mathrm{SO}}_{0}})C_{\mathrm{CA}}$$
and its solution can be obtained as
(9)$$C_{\mathrm{CA}}=\frac{(kC_{{\mathrm{CA}}_{0}}^{2}q-kC_{{\mathrm{SO}}_{0}}C_{{\mathrm{CA}}_{0}})e^{(kC_{{\mathrm{CA}}_{0}}-\frac{kC_{{\mathrm{SO}}_{0}}}{q})t}}{C_{{\mathrm{CA}}_{0}}kqe^{(kC_{{\mathrm{CA}}_{0}}-\frac{kC_{{\mathrm{SO}}_{0}}}{q})t}-kC_{{\mathrm{SO}}_{0}}}$$

It is obvious that the concentration of C_3_A is a function of time.

The variation of fractal dimensions can mainly be attributed to production of DEF. The relationship between the fractal dimension and DEF production was obtained according to the previous studies of Chen et al. [48]. As seen in Figure 12, the increase of fractal dimensions is approximately proportional to ettringite. Therefore, it is concluded that the increased rate of fractal dimensions (d*D*/d*t*) is proportional to the change rate of DEF concentrations, which can be expressed as
(10)$$\frac{dD}{dt}\propto \frac{dC_{\mathrm{DEF}}}{dt}$$
Based on Equations (5) and (6), the increase rate of DEF is proportional to the decreased rate of C_3_A, that is
(11)$$\frac{dC_{\mathrm{CA}}}{dt}=-\lambda \frac{dD}{dt}$$
where *λ* is a pending parameter.

By solving Equation (11), we have
(12)$$D=\frac{1}{\lambda }(C_{{\mathrm{CA}}_{0}}-C_{\mathrm{CA}})$$

Then, substituting Equation (12) into Equation (9), we obtain
(13)$$D=\frac{C_{{\mathrm{SO}}_{0}}C_{{\mathrm{CA}}_{0}}e^{(kC_{{\mathrm{CA}}_{0}}-\frac{kC_{{\mathrm{SO}}_{0}}}{q})t}-C_{{\mathrm{SO}}_{0}}C_{{\mathrm{CA}}_{0}}}{\lambda qC_{{\mathrm{CA}}_{0}}e^{(kC_{{\mathrm{CA}}_{0}}-\frac{kC_{{\mathrm{SO}}_{0}}}{q})t}-\lambda C_{{\mathrm{SO}}_{0}}}$$

Finally, Equation (13) can be simplified and written
(14)$$\begin{array}{l} D=\frac{E\left(e^{t/t_{u}}-1\right)}{e^{t/t_{u}}-F}\\ t_{u}=\frac{q}{k(qC_{{\mathrm{CA}}_{0}}-C_{{\mathrm{SO}}_{0}})}\\ E=\frac{C_{{\mathrm{SO}}_{0}}}{\lambda q}\\ F=\frac{C_{{\mathrm{SO}}_{0}}C_{{\mathrm{CA}}_{0}}}{q} \end{array}$$
where *t_u_*, *E* and *F* can be determined by fitting experimental results (see Figure 13), with the fitted values as listed in Table 3.

It is also seen from Figure 13 that the proposed model (Equation (14)) can describe the behavior of fractal dimensions of surface micro-cracks well. Actually, production and growth of DEF causes expansive stress in the concrete pore, which gives rise to micro-crack nucleation and growth. However, it should be mentioned that as the value of *F* in Equation (14) is small in three samples with different water-to-cement ratios, it can be ignored. Therefore, Equation (14) can be simplified and written as
(15)$$D=E\left(1-e^{-t/t_{u}}\right)$$
where the parameter *E* is the fractal dimension when the corrosion time tends to infinite, and it is directly proportional to the water-to-cement ratios of the samples. The attained values of fractal dimension *D* are all less than 2 when time tends to infinity. The main reason is that when the surface of the sample is cracked due to sulfate attack, the harmful ions slowly spread from the crack or pores to the inner of the sample. With the increase of corrosion time, the internal corrosion has been dominant. Hence, it is difficult for surface cracks to spread to the whole surface. The symbol *t_u_* is a characteristic time, at which the fractal dimension is equal to 0.632*E*. This characteristic time not only depends on the initial concentration of the sulfate solution, but also on the concentration of the initial un-hydrated C_3_A, as well as the chemical reaction constant. Although the number of samples in the tests was limited, with some discrete deviations, it is seen from the change of fractal dimensions in the crack propagation process of three different water-cement ratios that the final theoretical modeling results are consistent with the ones tested.

Here, let us respectively define the normalized fractal dimension and non-dimensional time as
(16)$$\begin{array}{l} D^{*}=D/E\\ t^{*}=t/t_{u} \end{array}$$

Then, substituting Equation (16) into Equation (15), we obtain
(17)$$D^{*}=1-e^{-t*}$$

That is, the normalized fractal dimension monotonically increases with the non-dimensional time. The relationship between *D** and *t** is universal for all samples with different water-to-cement ratios. This can be further confirmed by Figure 14, where all the re-scaled experimental data are well collapsed on a single curve.

## 5. Conclusions

In terms of the fractal concept, cracking patterns in concrete samples under sulfate attack were investigated. Based on experimental testing and theoretical analysis, the following remarks can be concluded:(1)Although the surface cracks are complex due to the arbitrary evolution of their lengths and directions, they can be described by a fractal dimension, which is an exponential function of non-dimensional time.(2)Fractal dimension exponentially increases with the corrosion time due to the chemical reaction speed.(3)The greater the values of water-to-cement ratios of concrete, the greater the damage degrees and fractal dimensions of the samples.(4)The chemo-mechanical method adopted in this paper can be applied to analyze the corrosion damage of concretes under sulfate attack, and it is also instructive to the evaluation of corrosion damage in other materials.

## Figures and Tables

**Figure 1 materials-12-02338-f001:**
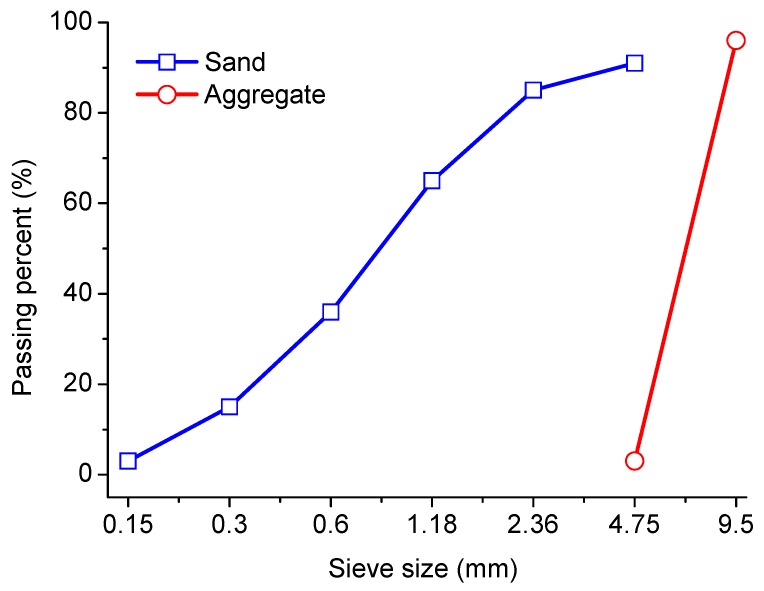
Particle size distributions of sand and aggregate.

**Figure 2 materials-12-02338-f002:**
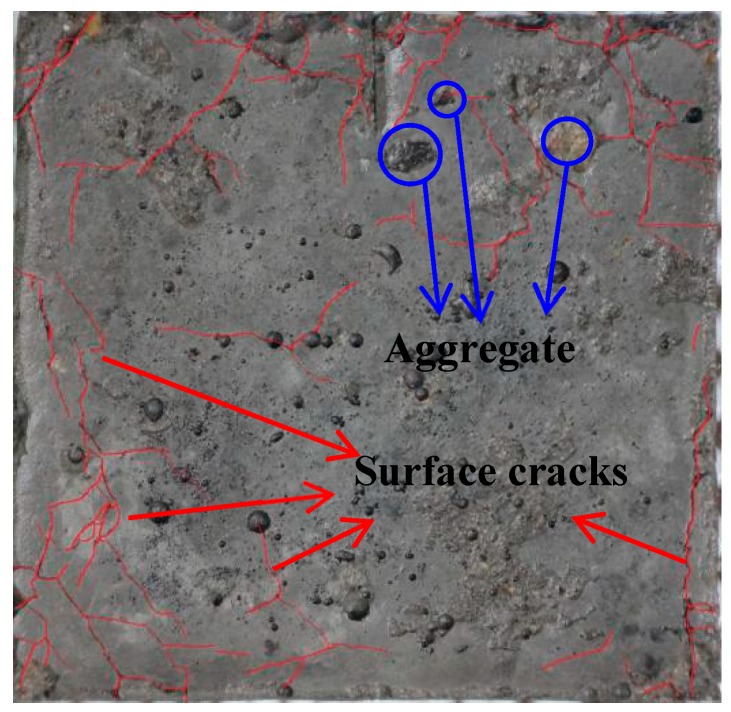
Distribution of aggregate on sample surface.

**Figure 3 materials-12-02338-f003:**
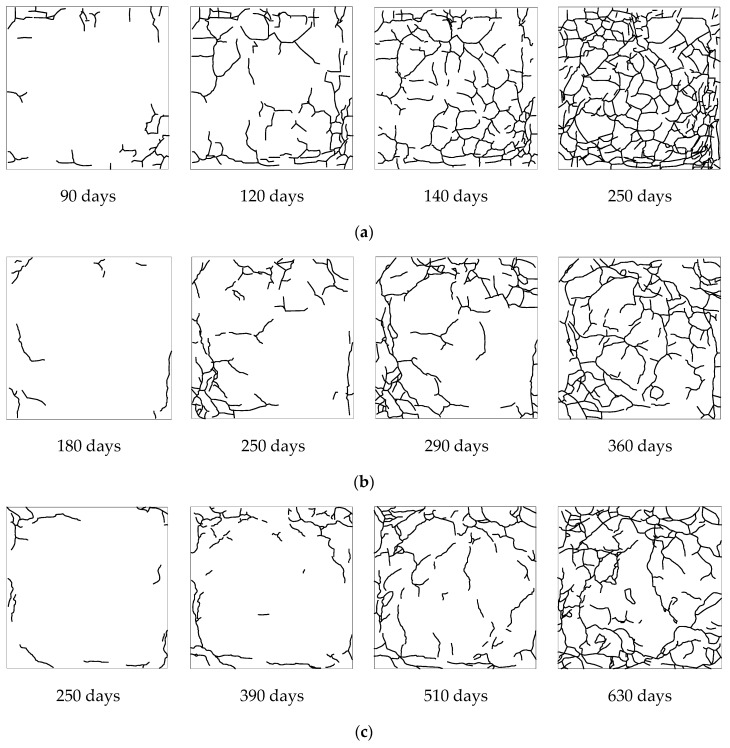
Evolution of surface cracking patterns in concretes with a water-to-cement ratio of (**a**) *w*/*c* = 0.65, (**b**) *w*/*c* = 0.55, and (**c**) *w*/*c* = 0.45.

**Figure 4 materials-12-02338-f004:**
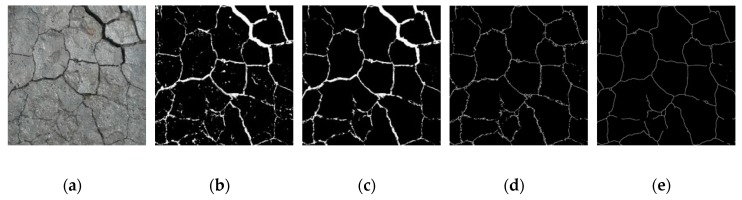
Schematic of digital image processing on a surface cracking pattern (**a**) including, (**b**) initial grey-scale image, (**c**) grey-scale image after de-noising, (**d**) skeletonization, and (**e**) removing noise.

**Figure 5 materials-12-02338-f005:**
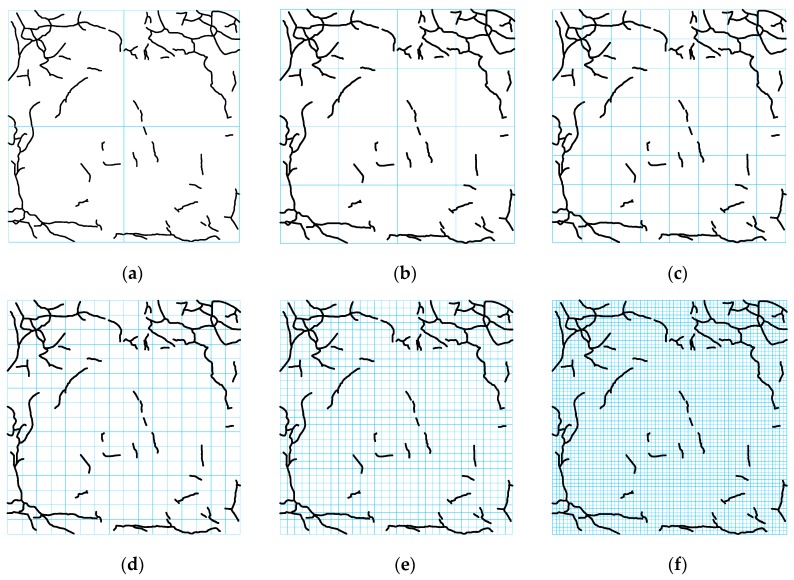
Methodology of calculating fractal dimension of surface cracks: (**a**) *r* = 1/2; (**b**) *r* = 1/4; (**c**) *r* = 1/8; (**d**) *r* = 1/16; (**e**) *r* = 1/32; and (**f**) *r* = 1/64.

**Figure 6 materials-12-02338-f006:**
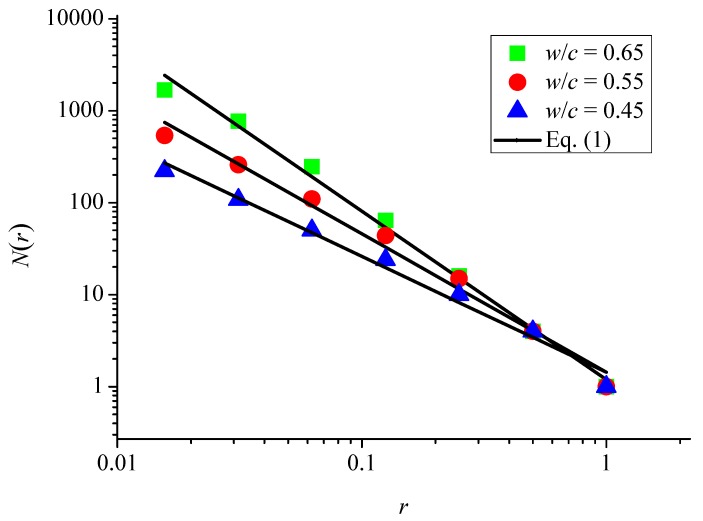
Fractal dimension of surface cracks in specimens at an aging time of 250 days.

**Figure 7 materials-12-02338-f007:**
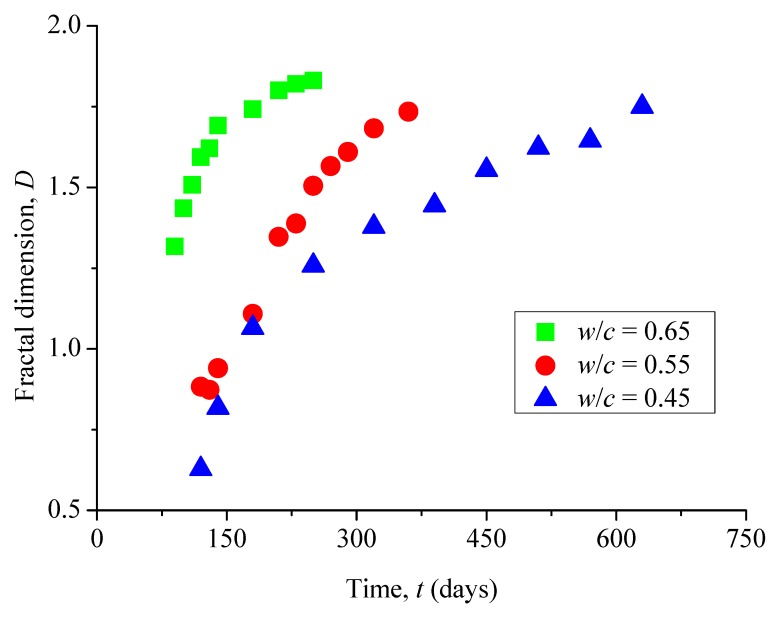
Evolution of fractal dimension of surface cracks with immersion time.

**Figure 8 materials-12-02338-f008:**
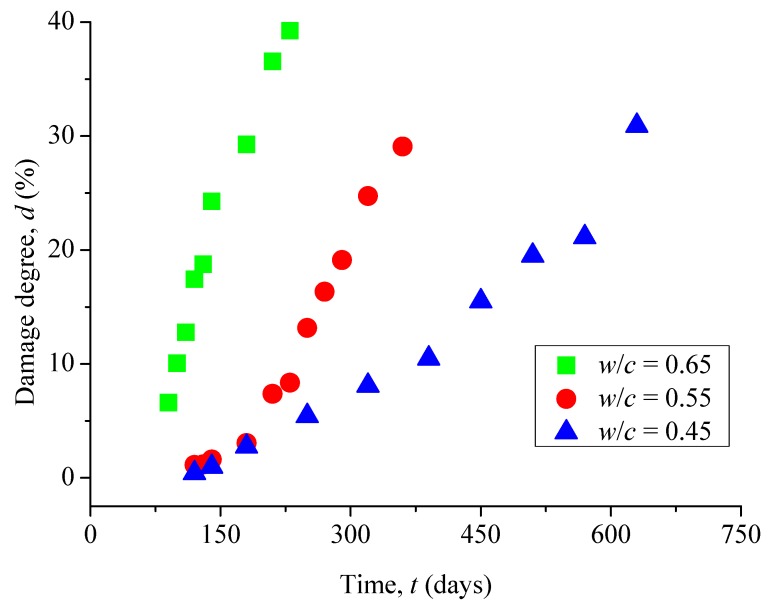
Evolution of the damage degree with immersion time.

**Figure 9 materials-12-02338-f009:**
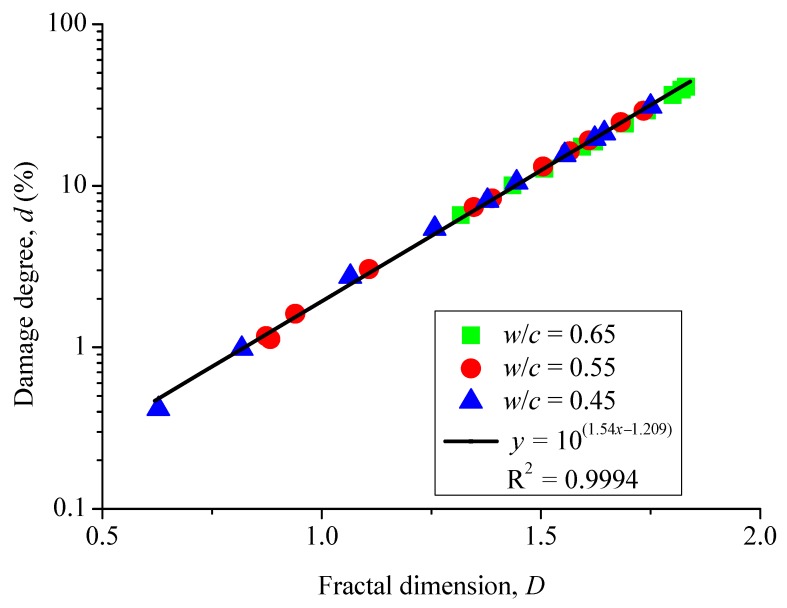
Fractal dimension versus damage degree.

**Figure 10 materials-12-02338-f010:**
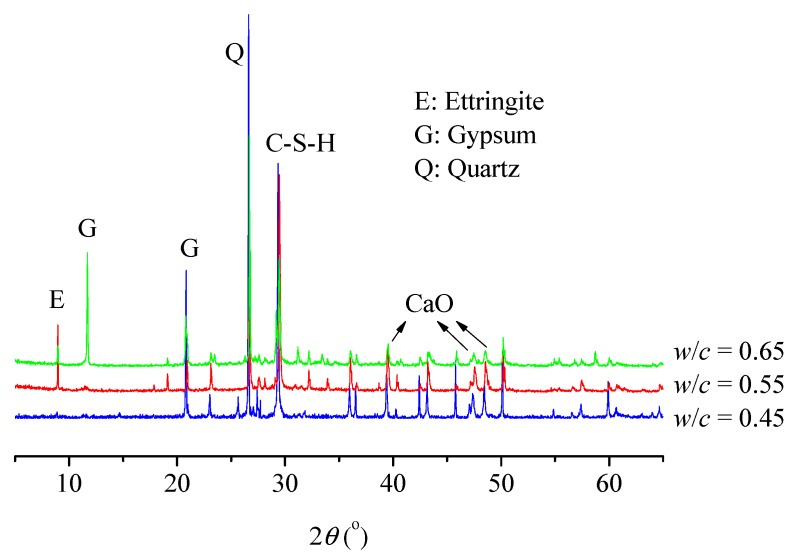
XRD patterns of sample powder for different specimens at a corrosion time of 250 days. Here, *θ* is the angle of incidence.

**Figure 11 materials-12-02338-f011:**
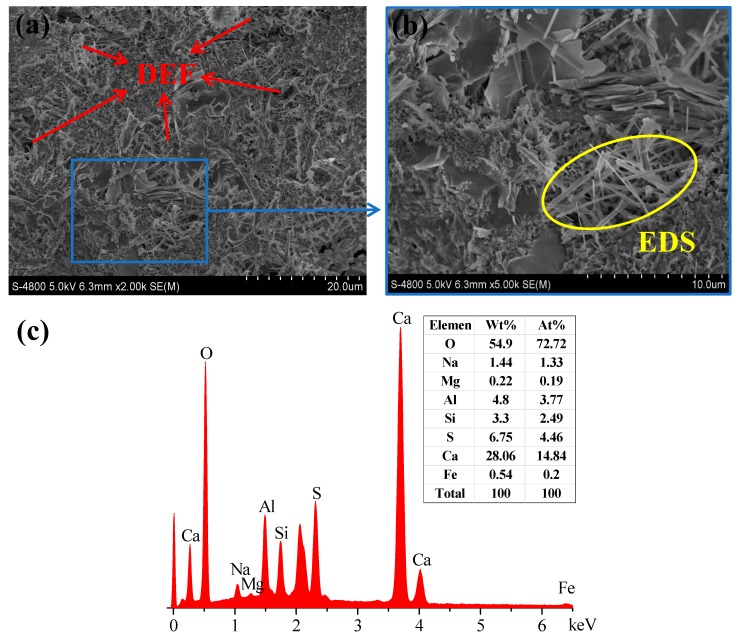
Photos of delayed ettringite formation (DEF) at a corrosion time of 600 days. (**a**) SEM image of DEF (2000× magnification), (**b**) SEM image of DEF (5000× magnification), and (**c**) energy spectrum analysis.

**Figure 12 materials-12-02338-f012:**
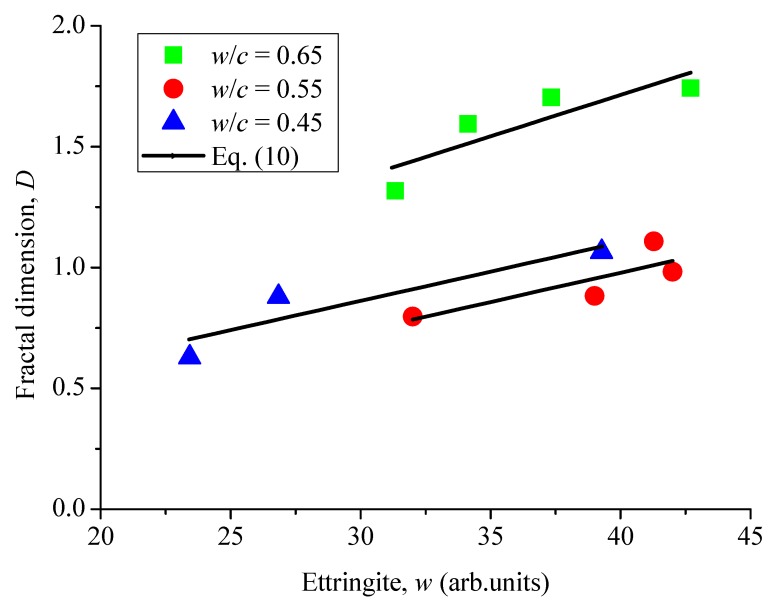
Fractal dimension of surface cracks versus the concentration of ettringite.

**Figure 13 materials-12-02338-f013:**
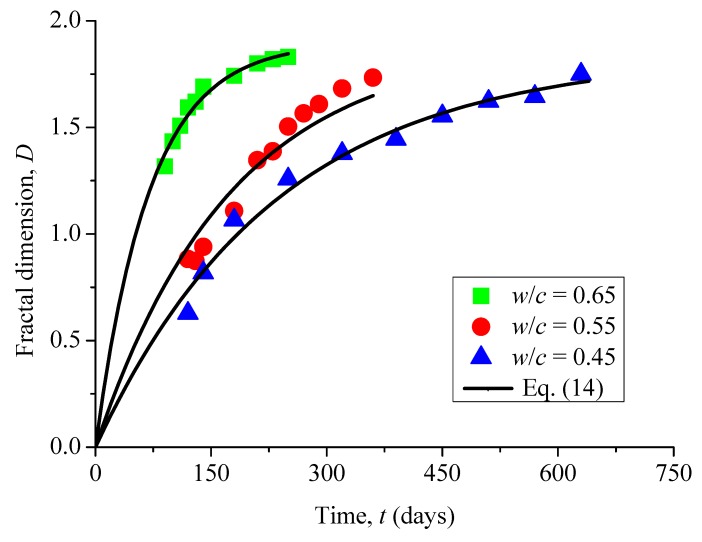
Results of fractal dimension as a function of immersion time.

**Figure 14 materials-12-02338-f014:**
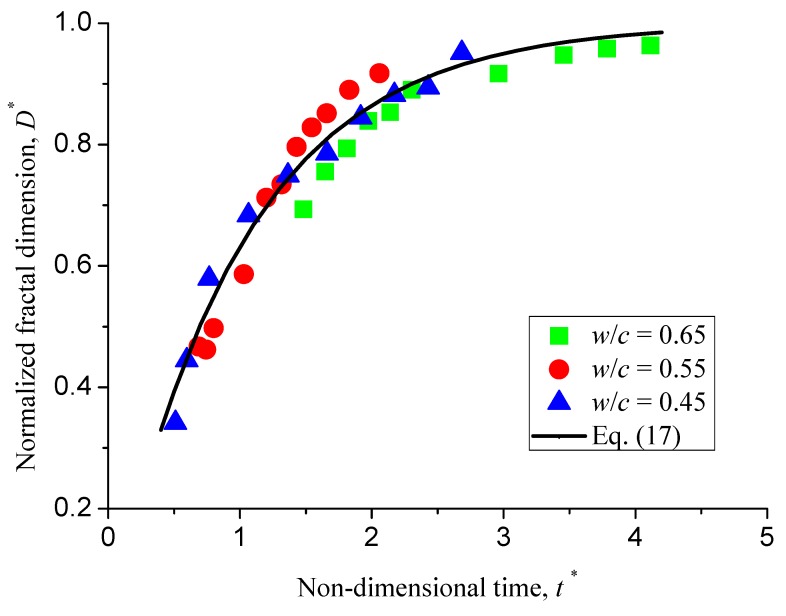
Normalized fractal dimension versus non-dimensional time for all samples with different water-to-cement ratios.

**Table 1 materials-12-02338-t001:** Chemical composition of cement in the unit of wt.%.

Al_2_O_3_	CaO	Fe_2_O_3_	K_2_O	MgO	Na_2_O	SO_3_	SiO_2_	TiO_2_	Loss
7.70	56.81	4.19	1.47	1.71	0.83	2.57	23.34	0.86	0.52

**Table 2 materials-12-02338-t002:** Mineral composition of cement in the unit of wt.%.

C_3_S	C_2_S	C_3_A	C_4_AF	Gypsum	SiO_2_	Calcite	Brushite
35.7	5.7	3.2	—	1.8	10.5	22.8	20.3

**Table 3 materials-12-02338-t003:** Magnitude of parameters *t_u_*, *E* and *F* for concrete samples with different water-to-cement ratios.

*w/c*	*t_u_*	*E*	*F*	R^2^
0.65	60.77	1.90	6.21 × 10^−10^	0.97
0.55	174.90	1.89	2.81 × 10^−14^	0.93
0.45	234.80	1.84	4.40 × 10^−12^	0.98

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
