# Peer review of "Fractal Cracking Patterns in Concretes Exposed to Sulfate Attack"

_materials, 2019, doi:10.3390/ma12142338_

Round 1

Reviewer 1 Report

The manuscript is related to study crack patterns of cement samples with different water-to-cement ratio using fractal analysis. Even, if the idea of the manuscript is interesting, the obtained results are not representative because the maximum size of the aggregate was 10 mm while the samples were 60x60x20 mm in size. Thus, the obtained results can not be accepted since it is well known that the presence of the aggregate affects the cracking behaviour of cement-based materials and the investigated samples were too small to state any scientific conclusions.

Reviewer 2 Report

Overall, a very interesting and fundamental approach to investigate the relationship between the fractal dimension of the surface cracks in concrete. The techniques which were utilized are acceptable and the EDS images are nice. however, Energy spectrum analysis (fig. 9b) can be presented better. 

Generally, the paper is acceptable. However, I was wondering how many SEM imaging was performed? Since replication is important for that conclusion 

Please, try to improve your conclusion both in shape and context. I believe you can add more to that. 

Reviewer 3 Report

The paper concerns the investigations about the characteristics of cracks patterns on the surface of a concrete, caused by the sulfate attack. The study is extensive and is well developed. The main structure of the paper is correct. The article has a scientific overtone. The big advantage of the work is that authors apart from the presentation of research results, they also proposed a theoretical model of the occurring dependencies. However, there are some shortcomings, which should be corrected before acceptance the paper for publication. Detailed comments are listed below:

1.      The paper should be checked by native English speaker in order to improve the style; there is also few syntax and grammar errors.

2.      Section 2.1 - If the authors performed such tests, please attach a sand and coarse aggregate particle size distribution curves.

3.      Section 2.2 - Why was considered the w/c = 0.65? This is a rather rare w/c ratio in practice. Please make a comment on this.

4.      Line 132-134 - To get the image for analysis, a very good approach is to use the technique of scanning the sample on a regular optical scanner in a very high resolution, e.g. 1200 DPI or 2400 DPI. Scanning in such a high resolution allows to detect also cracks on the surface of the material that are not visible to the naked eye. The authors may cite that, apart from the technique of taking photos of samples, there are other techniques for obtaining an image for analysis, e.g. the scanning. Examples of references to literature where this technique was used:

"Influence of specimen’s shape and size on the thermal cracks’ geometry of cement paste." Construction and Building Materials 189 (2018): 1155-1172.

"Evaluation of cracking patterns of cement paste containing polypropylene fibers." Composite Structures 220 (2019): 402-411.

5.      Figure 7 - Due to the universality of the relationship shown, it is worth calculating and giving in the article the equation of the regression curve for this relationship, together with the matching factors, e.g., coefficient of determination, random coefficient of variation, standard error of the estimation.

6.      Section 5 - This section could be more elaborate and more detailed.

Reviewer 4 Report

The manuscript deals with a topic of potential interest for journal readers. Experimental results are valuable, even if the number of tests seems quite limited to provide generalized conclusions (I suggest to comment on this in the manuscript). The normalized fractal dimension and a non-dimensional time allow to collapse the prediction equations in one single equation, and this is the main contribution of this work. Image processing of surface cracks represents a sound way to provide quantitative evaluations of a spread crack pattern. Introduction can be improved by adding other applications of Digital Image processing, like as those cited in

Bilotta, A., Ceroni, F., Lignola, G.P., Prota, A. ”Use of DIC technique for investigating the behaviour of FRCM materials for strengthening masonry elements” (2017) Composites Part B: Engineering, 129, pp. 251-270. DOI: 10.1016/j.compositesb.2017.05.075.

Tekieli, M., De Santis, S., de Felice, G., Kwiecień, A., Roscini, F. “Application of Digital Image Correlation to composite reinforcements testing” (2017) Composite Structures, 160, pp. 670-688. DOI: 10.1016/j.compstruct.2016.10.096.

Line 56, please revise “This also promote…”

Round 2

Reviewer 1 Report

The response is not satisfied. In my opinion the results are not representative for samples with size of 60x60x20 mm with the coarse aggregate. This aggregate influences significantly the cracking behaviour of samples. For this kind of size the samples without aggregate may be only investigated. 

Reviewer 3 Report

All suggestions and comments have been included. The paper has been improved. I accept the paper for publication.

Author Response

Thanks so much for your advice!